# Emergent Sasaki-Einstein geometry and AdS/CFT

Robert J. Berman [1✉], Tristan C. Collins [2✉] & Daniel Persson [1✉]

A central problem in any quantum theory of gravity is to explain the emergence of the classical spacetime geometry in some limit of a more fundamental, microscopic description of nature. The gauge/gravity-correspondence provides a framework in which this problem can, in principle, be addressed. This is a holographic correspondence which relates a supergravity theory in five-dimensional Anti-deSitter space to a strongly coupled superconformal gauge theory on its 4-dimensional flat Minkowski boundary. In particular, the classical geometry should therefore emerge from some quantum state of the dual gauge theory. Here we confirm this by showing how the classical metric emerges from a canonical state in the dual gauge theory. In particular, we obtain approximations to the Sasaki-Einstein metric underlying the supergravity geometry, in terms of an explicit integral formula involving the canonical quantum state in question. In the special case of toric quiver gauge theories we show that our results can be computationally simplified through a process of tropicalization.

[1] Department of Mathematical Sciences, Chalmers University of Technology, Gothenburg, Sweden. [2] Department of Mathematics, Massachusetts Institute of Technology, Cambridge, MA 02139, USA. ✉email: robertb@chalmers.se; tristanc@mit.edu; daniel.persson@chalmers.se

I t is expected that a quantum theory of gravity should be able to explain the emergence of the classical spacetime geometry in some limit of a more fundamental, microscopic description of Nature. The AdS/CFT-correspondence (or "gauge/gravity correspondence") introduced in ref. [1], provides a framework in which this problem can, in principle, be addressed. The AdS/CFT correspondence relates a supergravity theory in the five-dimensional Anti-deSitter space $AdS_5$ to a strongly coupled, rank $N$, superconformal gauge theory on the 4-dimensional flat boundary $\mathbb{R}^{3,1}$ of AdS$_5$. This is a holographic correspondence since $\mathbb{R}^{3,1}$ is the (conformal) boundary of AdS$_5$; hence it relates a gravitational theory in spacetime, to a conformal field theory (without gravity) on the boundary[2–4]. In particular, the classical geometry, i.e., the supergravity vacuum, should therefore emerge from a particular quantum state of the dual gauge theory. The main aim of our work is to make this precise by exhibiting a canonical (i.e., background independent) such quantum state $\Psi_N$ and by showing that the supergravity vacuum in question emerges from the probability amplitude of $\Psi_N$ in the t'Hooft limit where the rank $N$ of the gauge group tends to infinity.

In the general setting of minimal supersymmetry, the supergravity vacuum is encoded by a Sasaki–Einstein metric $g_M$ on a five-dimensional compact manifold $M$[5,6]. On the gauge theory side the $\mathcal{N} = 1$ superconformal symmetry is encoded by a complex cone $Y$ of six real dimensions. This means that $Y$ is a complex affine algebraic variety with a unique singular point $y_0$ (the tip of the cone), endowed with a repulsive holomorphic $\mathbb{R}_{>0}$-action, representing the conformal dilatation symmetry of the gauge theory. In the AdS/CFT correspondence the compact manifold $M$ on the supergravity side arises as to the base of the complex cone $Y$ of the gauge theory:

$$M := \left( Y - \{y_0\} \right) / \mathbb{R}_{>0}. \tag{1}$$

The complex cone $(Y, \mathbb{R}_{>0})$ comes with a canonical holomorphic three-form $\Omega$ which is homogeneous of degree 3. Such a space $Y$ is often called a Calabi–Yau cone in the literature and is usually endowed with a conical Calabi–Yau metric $g_Y$, i.e., a Ricci flat conical Kähler metric, as well as a radial coordinate $r$, arising as the distance to the vertex point $y_0$ with respect to the Calabi–Yau metric $g_Y$. However, in the background independent formalism that we shall stress, the complex cone $Y$ is not, a priori, endowed with any metric. This is crucial since our aim is to show how a metric emerges from the metric-independent BPS-sector of the gauge theory. Recall that BPS-states can be represented by holomorphic polynomial functions on the vacuum moduli space of the rank $N$ gauge theory, whose mesonic branch is given by the symmetric product $Y^N/S_N$, where $S_N$ denotes the symmetric group on $N$ elements. The vacuum moduli space can thus be described in purely complex algebro-geometric terms[7–9].

In this work, we show that by imposing all the manifest symmetries of the rank $N$ gauge theory one naturally arrives at a canonical quantum BPS-state $\Psi_N$. More precisely, $\Psi_N$ can be realized as a wave function on the vacuum moduli space $Y^N/S_N$ and its amplitude $|\Psi_N|^2$ induces a measure on $Y^N$:

$$|\Psi_N(y_1, ..., y_N)|^2 \left(\Omega \wedge \bar{\Omega}\right)^{\otimes N} \tag{2}$$

which is $S_N$–invariant and $\mathbb{R}_{>0}$–invariant along each factor of $Y^N$ (here $\bar{\Omega}$ denotes the conjugate of $\Omega$ multiplied by $-i$ so that $\Omega \wedge \bar{\Omega}$ is a volume form on $Y$). Somewhat surprisingly the state $\Psi_N$ does not appear to have been considered before in the literature. Its explicit expression is given in Section II C, where the relation to the BPS-sector of the gauge theory is also explained. By quotienting out the conformal $\mathbb{R}_{>0}$–symmetry, we arrive at a measure on $M^N/S_N$, which after normalization yields a canonical probability measure. Equivalently, we obtain a canonical ensemble of $N$ "point-particles" on $M$. We show that a Sasaki–Einstein

metric $g_M$ on $M$ emerges from the canonical ensemble in the large $N$-limit. In a little more detail, our construction goes as follows. First, the volume form $dV_M$ of $g_M$ emerges, after which the metric $g_M$ can be recovered from $dV_M$ in a standard manner (by simply differentiating $dV_M$ twice). In fact, in the course of this procedure an $N$-dependent radial function $r_N$ on $Y$ naturally appears in an intermediate step and it induces a "quantum correction" $g_M^{(N)}$ to the Sasaki–Einstein metric $g_M$ on $M$. In this way, a limiting radial coordinate $r$ on $Y$ emerges from the gauge theory as $N \to \infty$. From the perspective of the AdS/CFT correspondence, the radial coordinate $r$ on $Y$ corresponds to the radial coordinate on AdS$_5$ and thus our work reveals how the geometry of AdS$_5$, transversal to the conformal boundary $\mathbb{R}^{3,1}$, naturally emerges from the gauge theory. A "spin-off effect" of this procedure is that it also produces (by a kind of back-reaction) a conical Calabi–Yau metric on $Y$, namely the cone over $g_M$, whose distance to the vertex point $y_0$ is precisely $r$. It should be stressed that there are very few cases known where the Sasaki–Einstein metric $g_M$ on $M$ can be explicitly computed (but see ref. [10] for a notable family of exceptions). Thus an important feature of our construction is that it furnishes canonical approximations $g_M^{(N)}$ of Sasaki–Einstein metrics, encoded in terms of algebro-geometric data through an explicit integral formula. These can be numerically computed using Monte-Carlo methods, as detailed in section "Specialization to the toric case" for the toric case. Some relations to previous results and ideas for future work are discussed in section "Discussion".

## Results

**Background: AdS/CFT and BPS-states.** Recall that the low-energy dynamics of a general supersymmetric gauge theory is controlled by the moduli space of classical vacua $\mathcal{M}$. The space $\mathcal{M}$ may be defined as the critical points, modulo complex gauge equivalence, of the superpotential $W$ appearing in the (UV) Lagrangian of the gauge theory[11,12]. Moreover, the BPS-operators of the gauge theory, i.e., the local operators preserving half of the supercharges, may be represented by holomorphic, polynomial functions on $\mathcal{M}$. They thus form a ring known as the chiral ring of the gauge theory, which is graded by the $R$-charge. In the superconformal case, the BPS-operators can be viewed as states by the usual operator-state correspondence in CFT. The BPS-states are chiral primary states and saturate the BPS-bound,

$$\Delta = \frac{3}{2} R, \tag{3}$$

where $\Delta$ denotes the conformal dimension. As a consequence, the BPS-sector tends to be robust under non-perturbative corrections and can thus be used to probe the strong-coupling regime of the gauge theory.

In the setting of the AdS/CFT correspondence, the mesonic branch of the moduli space of vacua of the rank $N$ gauge theory is the symmetric product[8]

$$\mathcal{M}_N = \text{Sym}^N Y = Y^N / S_N, \tag{4}$$

where $Y$ is a complex cone. From the string theory perspective, this space parametrizes the transverse positions of $N$D3-branes inside $Y$. For example, in the maximally supersymmetric $SU(N)$-case, the superpotential $W(Z_1, Z_2, Z_3)$ is defined on 3 complex $N \times N$ matrices $Z_i$ (transforming in the adjoint representation) and $W = Tr(Z_1[Z_2, Z_3])$. Thus the mesonic vacuum moduli space $\mathcal{M}_N$ may be parametrized by the set $(\mathbb{C}^3)^N / S_N$ of joint eigenvalues of $(Z_1, Z_2, Z_3)$, showing that, indeed, $Y = \mathbb{C}^3$ in this case.

The mesonic BPS-sector in the maximally supersymmetric case is isomorphic to the ring of holomorphic, polynomial functions

on $(\mathbb{C}^3)^N/S_N$. Gauge theories with minimal supersymmetries may be constructed using quivers, encoding the matter content of the Lagrangian, as well as the superpotential[13]. In general, this is a highly non-trivial task, but our approach only requires that the corresponding moduli space of classical vacua, encoded by the complex cone $Y$, is given.

**Mathematical prerequisites: complex-geometric setup**. In this section we provide complex-geometric background, emphasizing a background-independent (i.e., metric-independent) perspective; see ref. [14] and the monograph[15] for the more standard metric-dependent point of view. Let $Y$ be a three-dimensional complex algebraic affine variety with an isolated singularity $y_0$. Concretely, $Y$ may be realized as the zero-locus of a collection of holomorphic polynomials on some complex space $\mathbb{C}^M$. Denote by $J$ the induced complex structure on the regular locus $Y - \{y_0\}$. Assume that $Y$ is endowed with a

- A repulsive holomorphic $\mathbb{R}_{>0}$-action that fixes $y_0$
- a holomorphic top-form $\Omega$, defined on the non-singular locus $Y - \{y_0\}$, which is homogeneous of degree 3 with respect to the $\mathbb{R}_{>0}$-action on $Y$.

Such a space $Y$ will here simply be called a complex cone, though it is often called a Calabi–Yau cone in the physics literature and an affine Gorenstein cone in the mathematics literature. The form $\Omega$ is uniquely determined up to a multiplicative constant and can often be written down explicitly.

The vector field on $Y$ generating the $\mathbb{R}_{>0}$-action will be called the dilatation vector field and denoted by $\delta$. Rotating $\delta$ with the complex structure $J$ on $Y$ yields another vector field that we shall denote by $\xi$:

$$\xi = J\delta. \tag{5}$$

The space $\mathcal{O}(Y)$ of holomorphic functions on $Y$ decomposes with respect to the $\mathbb{R}_{>0}$-action:

$$\mathcal{O}(Y) = \bigoplus_{k=0,1,\dots} \mathcal{O}_{\lambda_k}(Y), \qquad 0 = \lambda_0 \le \lambda_1 \le \cdots, \tag{6}$$

where $\mathcal{O}_{\lambda_k}(Y)$ is the vector space of holomorphic (polynomial) functions, which are homogeneous of degree $\lambda_k$ with respect to $\mathbb{R}_{>0}$. From the perspective of the underlying superconformal gauge theory, the infinitesimal $\mathbb{R}_{>0}$-action $\delta$ represents the conformal dilatation and $\frac{2}{3}\xi$ represents the R-symmetry (the factor 2/3 ensures that $\Omega$ has the same R-charge, 2, as the chiral superspace volume form $d^2\theta$, where $\theta$ denotes fermionic coordinates of positive chirality). Thus, Eq. (5) is the complex-geometric realization of the BPS-relation (3) and the summands $\mathcal{O}_{\lambda_k}(Y)$ in the decomposition (6) are thus BPS-states of dimension and R-charge equal to $\lambda_k$.

The base of a complex cone $Y$ is the compact five-dimensional manifold $M$ defined by

$$M := (Y - \{y_0\})/\mathbb{R}_{>0}. \tag{7}$$

Thus, $M$ is the base of the fibration $(Y - \{y_0\}) \to M$ obtained by quotienting out the $\mathbb{R}_{>0}$-action on $Y$. Since the vector field $\xi$ on $Y$ commutes with the generator of the $\mathbb{R}_{>0}$-action it induces a vector field on $M$ that we denote by the same symbol $\xi$—known as the Reeb vector field on $M$ in the mathematics literature. A metric $g_M$ on $M$ is said to define a Sasaki–Einstein metric on $(M, \xi)$ if $g_M$ has constant Ricci curvature, normalized so that it coincides with the Ricci curvature of the standard round metric on the unit-sphere, and is compatible, in a certain sense, with the complex structure on $Y$.

The compatibility in question can be formulated in various ways, but the crucial point for our purposes is that a Sasaki–Einstein metric $g_M$ can be explicitly recovered from its volume form $dV_{g_M}$ as follows. First observe that $dV_{g_M}$ induces a radial function, i.e., positive function $r$ on $Y$ which is one-homogeneous with respect to the $\mathbb{R}_{>0}$-action:

$$r := \left( \frac{\iota_\delta \Omega \wedge \bar{\Omega}}{dV_{g_M}} \right)^{1/6}, \tag{8}$$

where we have identified the volume form $dV_{g_M}$ on $M$ with its pull-back to $Y$. The metric $g_M$ on $M$ with volume form $dV_{g_M}$ is a Sasaki–Einstein metric iff the corresponding radial function $r$ on $Y$ solves the following PDE, after perhaps rescaling $r$,

$$\left( dd^c(r^2) \right)^3 = \Omega \wedge \bar{\Omega}, \;\; d^c := J^* d \tag{9}$$

on the regular locus $Y - \{y_0\}$ of $Y$. Here, $d$ denotes the exterior derivative and $d^c$ denotes its "rotation" by the complex structure $J$, so that $dd^c(r^2)$ defines a two-form on $Y$ and thus the three-fold exterior product $\left( dd^c(r^2) \right)^3$ defines a six-form on $Y$.

The PDE (9) is the celebrated Calabi–Yau equation on the complex cone $Y$; it is equivalent to the condition that the conical Kähler metric

$$g_Y := dd^c(r^2)(\cdot, J\cdot), \tag{10}$$

on $Y$ is a Calabi–Yau metric, i.e., the Ricci curvature of $g_Y$ vanishes. Finally, $g_M$ may be explicitly recovered by identifying $M$ with the level set $\{r = 1\}$ in $Y$ and letting $g_M$ be the restriction of $g_Y$ to the level set $\{r = 1\}$.

It is important to emphasize that, in general, the base $(M, \xi)$ of a complex cone $(Y, \mathbb{R}_{>0})$ may not admit a Sasaki–Einstein metric[9,14,16]. Equivalently, this means that a complex cone may not admit a conical Calabi–Yau metric and hence no radial solution $r$ to the Calabi–Yau equation (9). Indeed, by[16] there exists a Sasaki–Einstein metric on $(M, \xi)$ iff the complex cone $(Y, \mathbb{R}_{>0})$ is K-stable. This is a purely algebro-geometric condition.

As shown in ref. [17] the K-stability of $Y$ can be viewed as a generalized form of the maximization condition for the $a$-central charge of the SCFT. This means that, in general, there are obstructions to the existence of a SCFT with a given $\mathbb{R}_{>0}$-graded mesonic chiral ring $\mathcal{O}(Y^N/S_N)$. In the present approach, a different, but conjecturally equivalent, stability type condition naturally appears, which is a variant of the notion of Gibbs stability introduced in the context of Fano manifolds in ref. [18].

**Main results: emergent geometry**. According to the AdS/CFT-correspondence the classical supergravity vacuum geometry in $AdS_5$ should emerge from some limit of a quantum state in the dual CFT gauge theory on the boundary. Concretely, in the present setting the non-trivial part of the supergravity vacuum in question is encoded by a Sasaki–Einstein metric $g_M$ on the internal compact space $M$, corresponding to the base of the complex cone $Y$ of the dual gauge theory[6]. Hence, we want to show that the Sasaki–Einstein metric $g_M$ on $M$ emerges in a certain "large $N$-limit" of a specific (and background free) quantum BPS-state $\Psi_N$ in the dual-rank $N$ gauge theory.

First, recall that $Y$ is endowed with a holomorphic top-form $\Omega$ and hence one can endow the mesonic classical vacuum moduli space $Y^N/S_N$ with the volume form $(\Omega \wedge \bar{\Omega})^{\otimes N}$. Let $\psi_1, \dots, \psi_{N_k}$ be a maximal number of linearly independent mesonic BPS-states for the rank 1 gauge theory with the same R-charge $\lambda_k$. In other words, $\psi_1, \dots, \psi_{N_k}$ form a basis in the space $\mathcal{O}_{\lambda_k}(Y)$ of holomorphic functions on $Y$ with the same charge $\lambda_k$. Denote

by $\Psi_{\det}$ the corresponding Slater determinant, i.e., the totally antisymmetric holomorphic function on $Y^{N_k}$ given by

$$\Psi_{\det}(y_1, \dots, y_{N_k}) := \sum_{\sigma \in S_{N_k}} (-1)^{|\sigma|} \psi_{\sigma(1)}(y_1) \cdots \psi_{\sigma(N_k)}(y_{N_k}). \quad (11)$$

The function $\Psi_{\det}$ is independent of the choice of bases in $\mathcal{O}_{\lambda_k}(Y)$ up to an overall multiplicative constant. It thus defines a baryonic BPS-state in the rank $N_k$ gauge theory[8,19].

We aim to construct a symmetric measure on the mesonic classical vacuum moduli space $Y^{N_k}/S_{N_k}$, which is invariant under the conformal $\mathbb{R}_{>0}$−action on each factor. One might be tempted to try with the density $|\Psi_{\det}|^2$, but this is unfortunately not $(\mathbb{R}_{>0})^{N_k}$-invariant. The resolution is to simply take a suitable fractional power of the Slater determinant, namely

$$\Psi_{N_k} := \Psi_{\det}^{-3/\lambda_k}. \quad (12)$$

Then, an $(\mathbb{R}_{>0})^{N_k}$-invariant measure on $Y^N/S_N$ is given by the combination

$$|\Psi_{N_k}(y_1, \cdots, y_{N_k})|^2 \, (\Omega \wedge \overline{\Omega})^{\otimes N_k}. \quad (13)$$

This measure is both invariant under the $S_{N_k}$ permutation symmetry and the conformal $\mathbb{R}_{>0}$-symmetry, as well as the R-symmetry. However, since $Y$ is non-compact, the integral of this measure over $Y^{N_k}/S_{N_k}$ diverges. To circumvent this problem, we simply quotient by the $(\mathbb{R}_{>0})^{N_k}$-action to get an induced measure on the compact space $M^{N_k}/S_{N_k}$. To be specific, contracting the top-form $\Omega \wedge \overline{\Omega}$ on $Y$ with the dilatation vector field $\delta$ yields a 5-form $\iota_\delta(\Omega \wedge \overline{\Omega})$ on $Y$. Thus, the $(\mathbb{R}_{>0})^{N_k}$−invariant form

$$|\Psi_{N_k}(y_1, \cdots, y_{N_k})|^2 (\iota_\delta \Omega \wedge \overline{\Omega})^{\otimes N_k} \quad (14)$$

maybe identified with a measure on the compact quotient space $M^{N_k}/S_{N_k}$. Finally, the canonical probability measure $d\mathbb{P}_{N_k}$ on $M^{N_k}/S_{N_k}$ is defined by

$$d\mathbb{P}_{N_k} := \frac{1}{\mathcal{Z}_{N_k}} |\Psi_{N_k}|^2 (\iota_\delta(\Omega \wedge \overline{\Omega}))^{\otimes N_k}, \quad (15)$$

where

$$\mathcal{Z}_{N_k} := \int_{M^{N_k}/S_{N_k}} |\Psi_{N_k}|^2 (\iota_\delta(\Omega \wedge \overline{\Omega}))^{\otimes N_k}. \quad (16)$$

Note that in (15) we have made the implicit assumption that $\mathcal{Z}_{N_k}$ is finite. Since $|\Psi_{N_k}|^2$ blows-up along a hypersurface in $Y^{N_k}/S_{N_k}$, namely the zero-locus of the Slater determinant $\Psi_{\det}$, this is actually a very non-trivial condition. We interpret it as a consistency condition (which turns out to be related to the mathematical notions of K-stability and Gibbs stability).

Since $\Psi_N$ is holomorphic away from its singularity locus in $Y^N/S_N$ (the vanishing locus of $\Psi_{\det}$) the state $\Psi_N$ can be viewed as a bound state of BPS-states in the rank $N$ gauge theory.

We emphasize that by "canonical" we mean that the definition of $d\mathbb{P}_{N_k}$ is background independent, in the sense that it does not depend on any underlying metric on $Y$ or $M$. It only depends on the complex structure $J$ on the classical vacuum moduli space and the $\mathbb{R}_{>0}$-action and thus only on the superconformal symmetry of the rank $N$-gauge theory. This is a crucial point for describing the emergence of the classical Sasaki–Einstein metric $g_M$ on $M$, which is our main focus.

We have also restricted the values of the rank $N$ to be a sequence of integers

$$N_k := \dim \mathcal{O}_{\lambda_k}(Y), \quad (17)$$

i.e., the multiplicity of the R-charge $\lambda_k$. This can be seen as a quantization condition. As is well-known, in the quasi-regular case (discussed in the next section) $N_k$ is a polynomial in $k$ of the form,

$$N_k = \frac{\lambda_k^2}{2} V + \mathcal{O}(k^1), \quad \lambda_k \sim k, \quad k \to \infty \quad (18)$$

where the positive number $V$ is an algebraic invariant of the complex cone $(Y, \mathbb{R}_{>0})$, known as its volume[9,14,16].

Assume, for simplicity, that the complex cone $(Y, \mathbb{R}_{>0})$ associated to the gauge theory is quasi-regular. A complex cone $(Y, \mathbb{R}_{>0})$ is quasi-regular if (up to a rescaling) the $\mathbb{R}_{>0}$−action on $Y$ can be complexified to a holomorphic $\mathbb{C}^\times$−action. Denote by $d\mathbb{P}_N^{(1)}$ the probability measure on $M$ defined as the $1 - \text{point}$ correlation measure of the canonical ensemble $(d\mathbb{P}_N, M^N/S_N)$ introduced in the previous section. In other words, $d\mathbb{P}_N^{(1)}$ is obtained by "integrating out" all but one of the factors of $M^N$:

$$d\mathbb{P}_N^{(1)}(y) = \frac{1}{\mathcal{Z}_N} \int_{M^{N-1}/S_{N-1}} |\Psi_{N_k}(y, y_2, ..., y_N)|^2 \times (\iota_\delta(\Omega \wedge \overline{\Omega}))^{\otimes N-1}. \quad (19)$$

Our main conjecture can now be stated as follows:

*Conjecture A:* Assume that the canonical ensemble $(d\mathbb{P}_N, M^N/S_N)$ is well-defined, i.e., that $\mathcal{Z}_N < \infty$. Then

(i)  The one-point correlation measure $d\mathbb{P}_N^{(1)}$ converges, as $N \to \infty$, to the volume form $dV_M$ of a Sasaki–Einstein metric $g_M$ on $(M, \xi)$, normalized to have unit-volume,

$$\lim_{N \to \infty} d\mathbb{P}_N^{(1)} = dV_M; \quad (20)$$

(ii)  The sequence of radial functions

$$r_N := \left( \frac{d\mathbb{P}_N^{(1)}}{\iota_\delta(\Omega \wedge \overline{\Omega})} \right)^{-1/6} \quad (21)$$

on the complex cone $(Y, R_{>0})$ converges, as $N \to \infty$, toward the radial function $r$ of the Calabi–Yau metric $g_Y$ on $Y$ corresponding to the Sasaki–Einstein metric $g_M$ on $(M, \xi)$.

(iii)  Conversely, if there exists a unique Sasaki–Einstein metric $g_M$ on $M$, then $\mathcal{Z}_N < \infty$.

We emphasize that for finite $N$, the radial function $r_N$ yields an explicit approximation $g_M^{(N)}$ to the Sasaki–Einstein metric $g_M$ on $M$ by identifying $M$ with the level set $\{r_N = 1\}$ and setting

$$g_M^{(N)} = dd^c(r_N^2)_{|M}(\cdot, J\cdot) \quad (22)$$

(c.f. Eq. (10)). Hence, part (ii) of the conjecture is equivalent to

$$\lim_{N \to \infty} g_M^{(N)} = g_M, \quad (23)$$

which is the sought-after emergence of the Sasaki–Einstein metric on $M$ in the large $N$ limit.

To be mathematically precise, the convergence statements in the conjecture are supposed to hold in the standard weak topologies. In fact, we make the stronger conjecture that the random measure $N^{-1} \sum_{i=1}^N \delta_{x_i}$ on the canonical ensemble converges in law toward the deterministic measure $dV_M$. Below we will prove a $\beta$-deformed version of this conjecture.

We first introduce a real-analytic family of probability measures $d\mathbb{P}_{N,\beta}$ on $M^N/S_N$, defined for a real parameter $\beta$, such that $d\mathbb{P}_{N,\beta}$ coincides with $d\mathbb{P}_N$ for $\beta = -1$, if $\mathcal{Z}_N < \infty$. To this end, fix a background radial function $r_0$ on $Y$. We can then identify the base $M := (Y - \{y_0\})/\mathbb{R}_{>0}$ of the cone $Y$ with the

level set $\{r_0 = 1\}$ in $Y$ and define $d\mathbb{P}_{N,\beta}$ as follows:

$$d\mathbb{P}_{N,\beta} := \frac{1}{\mathscr{Z}_{N,\beta}} \left| \Psi_{\det}(y_1, y_2, ..., y_N) \right|^{2 \left| \frac{3\beta}{\lambda_k} \right|} dV_0^{\otimes N}, \qquad (24)$$

where $dV_0$ denotes the volume form on $M$ obtained by restricting the five-form $\iota_\delta \Omega \wedge \bar{\Omega}$ to the level set $\{r_0 = 1\}$ and $\mathscr{Z}_{N,\beta}$ is the corresponding normalization constant (recall that $N$ is the multiplicity of the charge $\lambda_k$). The parameter $\beta$ can be viewed as a regularization parameter, since $\mathscr{Z}_{N,\beta}$ is automatically finite when $\beta > 0$ (or slightly negative). However, it should be stressed that it is only in the canonical case $\beta = -1$ that the probability measure (24) is independent of the choice of radial function $r_0$. Let $r_{N,\beta}$ be the radial function on $Y$ defined by

$$r_{N,\beta} := \left( \frac{d\mathbb{P}_{N,\beta}^{(1)}}{dV_0} \right)^{1/6\beta} r_0 \qquad (25)$$

(coinciding with $r_N$ when $\beta = -1$) and denote by $g_{M,\beta}^{(N)}$ the corresponding metric on $M$, obtained by replacing the radial function $r_N$ in Eq. (22) with $r_{N,\beta}$. We then have:

*Theorem B:* For each $\beta > 0$, there exists

(i) a volume form $\mu_\beta$ on $M$ such that

$$\lim_{N \to \infty} d\mathbb{P}_{N,\beta}^{(1)} = \mu_\beta,$$

(ii) a radial function $r_\beta$ on $Y$ such that

$$\lim_{N \to \infty} r_{N,\beta} = r_\beta,$$

(iii) and a Sasaki metric $g_{M,\beta}$ on $M$ such that

$$\lim_{N \to \infty} g_{M,\beta}^{(N)} = g_{N,\beta}.$$

Moreover, if $(M, \xi)$ admits a Sasaki–Einstein metric, then $r_\beta$ and $g_{M,\beta}$ extend real-analytically to $[-1, \infty]$ and setting $\beta = -1$ yields a Sasaki–Einstein metric $g_M$ on $M$.

The proof is given in §IV. In the course of the proof, we will show that the square of the limiting radial function $r_\beta$ is the unique conical Kähler potential on $Y$ solving the following PDE on $Y - \{y_0\}$:

$$(dd^c r_\beta^2)^3 = \left( \frac{r_\beta^2}{r_0^2} \right)^{3(\beta+1)} \Omega \wedge \bar{\Omega}. \qquad (26)$$

In particular, for $\beta = -1$ this is indeed the Calabi–Yau equation (9) for the radial function $r$ corresponding to a Sasaki–Einstein metric $g_M$ on $M$.

**Finiteness properties of the normalizing constant.** Loosely speaking, Theorem B thus shows that Conjecture A holds after the analytic continuation. More precisely, it shows that Conjecture A holds under the assumption that the order of taking the limits $N_k \to \infty$ and $\beta \to -1$ may be interchanged. By a physics level of rigor Conjecture A may thus be considered as established. However, we do expect that the introduction of the parameter $\beta$ is not needed and, in particular, that $\mathscr{Z}_N < \infty$ if and only if $(M, \xi)$ admits a unique Sasaki–Einstein metric. In the case when the Sasaki–Einstein metric is not unique, i.e., when the Lie algebra $\mathfrak{g}(Y, \xi)$ of the automorphism group of $(Y, \xi)$ is non-trivial[20], we conjecture that $\mathscr{Z}_{N,\beta} < \infty$ for any $\beta > -1$ when $N$ is sufficiently large. The "only if direction" can be deduced from recent mathematical results in complex geometry for complex cones $Y$ of any dimension, as will be shown elsewhere. Proving the remaining direction appears, however, to be very challenging, except in the case when $Y$ has complex dimension two, where a direct proof of Conjecture A can be given.

For example, when $Y = \mathbb{C}^2$ realizing $M$ as the Hopf fibration over the two-sphere $S^2$ and factorizing $\Psi_{\det}(x_1, x_2, ..., x_N)$ reveals that $\mathscr{Z}_{N,\beta}$ can be expressed as the configurational partition function for $N$ particles on $S^2$ interacting by the 2D-gravitational force with a mean-field scaling:

$$\mathscr{Z}_{N,\beta} = C_N \int_{(S^2)^N} \prod_{1 \le i \ne j \le N} \| x_i - x_j \|_{\mathbb{R}^3}^{\frac{2\beta}{N-1}} dA^{\otimes N}, \qquad (27)$$

expressed in terms of the restriction to $S^2$ of the Euclidean norm on $\mathbb{R}^3$ (c.f. Eq. (36)). Applying the arithmetic-geometric means inequality reveals that the integral is finite precisely when $\beta > -(1 - 1/N)$. A similar argument applies to any $Y$ of complex dimension two, using that $Y$ is a Kleinian singularity, i.e., $Y = \mathbb{C}^2/G$ for a finite subgroup $G$ of $SU(2)$ and thus that $M$ is a Seifert fibration over $S^2$, branched over three points (in this case $\mathscr{Z}_{N,\beta} < \infty$ for $\beta = -1$-when $N$ is sufficiently large-since $\mathfrak{g}(Y, \xi)$ is trivial; details will appear elsewhere). For higher-dimensional $Y$ the Slater determinant $\Psi_{\det}(x_1, x_2, ..., x_N)$ can not, however, be factorized. But a condition ensuring that our canonical partition function $\mathscr{Z}_N$ is finite for any $N$ is that $Y$ is an exceptional singularity[21,22]. For example, there are exactly (up to conjugation) five cases of exceptional (non-isolated) singularities of the form $Y = \mathbb{C}^3/G$, for $G$ a finite subgroup of $SL(\mathbb{C}, 3)$; notably Klein's simple group of order 168, $PSL(2, 7)$,[21], Cor 3.15. See ref. [23] for the construction of the matter content and gauge groups of the corresponding gauge theories; the quiver graphs for the five "exceptional" groups $G$ appear in figure 5 in refs. [23].

Moreover, a list of three-dimensional exceptional quasi-homogeneous (isolated) hypersurface singularities in $\mathbb{C}^4$ is given in ref. [22], Cor 1.1. Consider for example the case when $Y$ is a Briskorn–Pham singularity:

$$\{Z_0^{a_0} + Z_1^{a_1} + Z_2^{a_2} + Z_3^{a_3} = 0\} \subset \mathbb{C}^4, \qquad (28)$$

endowed with the diagonal $\mathbb{C}^*$−action with weights proportional to $a_i^{-1}$. If the powers $a_i$ are coprime and taken in increasing order, then $Y$ is an exceptional Gorenstein affine variety iff $1 < \sum_{i=0}^3 a_i^{-1} < 1 + a_3^{-1}$. This condition is, for example, satisfied for the powers $(2, 3, 7, 11)$ and is, in fact, equivalent to the condition that $\mathscr{Z}_N < \infty$ for any $N$. It also implies that the base of $Y$ admits a Sasaki–Einstein metric. However, in Conjecture A we only demand that $\mathscr{Z}_N < \infty$ for $N$ sufficiently large.

**Discussion**

Let us conclude by briefly mentioning some relations to previous work. First of all, our work is very much in the spirit of the program for emergent geometry in AdS/CFT initiated by Berenstein[24,25]. The main new feature in our work is the appearance of a negative and fractional power of the Slater determinant in the definition of the state $\Psi_N$ (see Eq. (12)) and its $\beta$-deformation. This is crucial in order to obtain background independence and to see the emergence of the spacetime metric, as explained in Section II C.

As explained at the end of section "Methods", our approach builds on the probabilistic approach to Kähler–Einstein metrics on Fano manifolds introduced in refs. [18,26,27], which, in turn, is motivated by the Yau–Tian–Donaldson (YTD) conjecture for Fano manifolds. A different connection between the YTD conjecture and AdS/CFT was first exhibited in ref. [17] (compare the discussion on stability in section "Mathematical prerequisites: complex geometric setup").

Finally, we note that our canonical ensemble on $M$ may be viewed as an ensemble of $N$ dual giant gravitons[28], as will be elaborated on in a separate publication.

## Methods

**Proof of Theorem B.** We will show how to deduce the theorem from the results in refs. [27],[29] concerning a probabilistic approach to Kähler–Einstein metrics on Fano manifolds. We thus start with some well-known geometric preparations to realize $M_0$ as a fibration over a Fano manifold (orbifold); see ref. [30], Section 2.3. First assume that $\xi$ is a regular Reeb vector field. This means that the orbits of the complexification of $\xi$ coincide with the orbits of a $\mathbb{C}^*$–action on $Y$ without fixed points on $Y^* := Y - \{y_0\}$. The corresponding compact complex manifold $X := Y^*/\mathbb{C}^*$ is a Fano manifold, i.e., the dual $K_X^*$ of its canonical line bundle $K_X$ is ample. The natural projection from $Y$ to $X$ realizes $Y^*$ as the total space of the $q$th tensor power $K_X^{\otimes q} \to X$ for some rational positive number $q$, when the zero-section has been removed (and $Y$ gets identified with the variety obtained by blowing down the zero-section). For example, in the cases when $Y$ is $\mathbb{C}^3$ and the conifold one gets $X = \mathbb{P}^2$ and $X = \mathbb{P}^1 \times \mathbb{P}^1$ with $q = 1/3$ and $q = 1/2$, respectively. In other words, denoting by $L$ the ample line bundle $(K_X^*)^{\otimes q}$, we may identify $Y$ with the total space of the fibration $L^* \to X$, with the zero-section deleted. The fixed radial function $r_0$ on $Y$ corresponds to a Hermitian metric $\|\cdot\|$ on $L^*$ and the induced quotient fibration $M_0 \to X$ realizes $M_0$ is a principal $U(1)$-bundle over $X$, namely the unit-circle bundle of $(L^*, \|\cdot\|)$:

$$\begin{array}{ccc} M_0 & \hookrightarrow & L^*(= Y^*) \\ & \searrow & \downarrow \\ & & X \end{array} \qquad (29)$$

where

$$M_0 = \{r_0 = 1\} = \{\|\cdot\| = 1\}. \qquad (30)$$

As a consequence, there is a one-to-one correspondence between the space $\mathcal{P}(M_0)^\xi$ of $\xi$-invariant probability measures $\mu$ on $M_0$ and the space $\mathcal{P}(X)$ of probability measure $\nu$ on $X$:

$$\mu = \nu \otimes d\theta, \qquad (31)$$

expressing $\mu$ as the fiber product of $\nu$ with the $\xi$-invariant probability measures $d\theta$ defined on the fibers of the fibration $M_0 \to X$. In other words, $\nu$ is proportional to the contraction of $\mu$ with $\xi$, descended to $X$. Introducing local holomorphic coordinates $z$ on $X$ and locally trivializing $L^*$ with the holomorphic section $(dz)^{\otimes q}$ of $K_X^{\otimes q}$ we may locally express

$$\Omega = dz \wedge d(w^{1/q}), \quad r_0^2 = \left(|w|^2 e^{\phi_0(z)}\right)^{1/3q}, \qquad (32)$$

where $w$ is a local holomorphic coordinate along the fibers of $L^*$ and $e^{\phi_0(z)}$ denotes the squared norm of $(dz)^{\otimes q}$, i.e., $e^{\phi_0(z)} = \| (dz)^{\otimes q} \|^2$. The local formula for $\Omega$ follows from the observation that $dz \wedge d(w^{1/q})$ glues to define an equivariant global holomorphic three-form on $Y^*$. The appearance of the power $1/3q$ in the formula for $r_0^2$ then follows from the relation $\xi = 3q\xi_{L^*}$, where $\xi_{L^*}$ is the standard $U(1)$-action along the fibers of $L^*$ (satisfying $\xi_{L^*} w = iw$), resulting from the normalization condition $\xi\Omega = 3i\Omega$. Since the weight space $H_{\lambda_k}(Y)$ may be identified with the space $H^0(X, L^{\otimes k})$ of holomorphic sections of the $k$th tensor power of the holomorphic line bundle $L \to X$ it also follows that $\lambda_k = 3qk$. Concretely, the identification in question is obtained by noting that an element $\Psi$ in $H_{\lambda_k}(Y)$ may be locally expressed as $\Psi(z, w) = f_k(z)w^k$ for a local holomorphic function $f_k(z)$, globally transforming as a holomorphic section of $L^{\otimes k} \to X$. In particular,

$$\Psi_{\det} = f_{\det}(z_1, ..., z_N)w_1^k \cdots w_N^k \qquad (33)$$

where the local holomorphic function $f_{\det}(z_1, ..., z_N)$ on $X^N$ transforms as a holomorphic section of $(L^{\otimes k})^{\boxplus N} \to X^N$, namely as the Slater determinant of $H^0(X, L^{\otimes k})$. Moreover, by Eq. (32) we have

$$\begin{aligned} |\Psi_{\det}|^2_{|M_0^N} &= |f_{\det}(z_1, ..., z_N)|^2 e^{-k\phi_0(z_1)} \cdots e^{-k\phi_0(z_N)} \\ &=: \|f_{\det}\|^2, \end{aligned} \qquad (34)$$

using, in the last equality, the induced Hermitian metric $\|\cdot\|$ on $(L^{\otimes k})^{\boxplus N_k} \to X^N$. With these preparations in place, we can thus express

$$d\mathbb{P}_{N,\beta}^{(1)} = \nu_{N,\beta} \otimes d\theta^{\otimes N}, \qquad (35)$$

where $\nu_{N,\beta}$ is the probability measure on $X^N$ defined by

$$\nu_{N,\beta} = \frac{\|f_{\det}\|^{\frac{2\beta}{kq}} dV_X^{\otimes N}}{\int_{X^N} \|f_{\det}\|^{\frac{2\beta}{kq}} dV_X^{\otimes N}}, \qquad (36)$$

where the volume form $dV_X$ on $X$ corresponds to the volume form $dV_0$ on $M_0$ under the correspondence (31). The probability measure $\nu_{N,\beta}$ on $X^N$ is precisely the probability measure defined by the "temperature deformed" determinantal point process on $X$ introduced in ref. [27], associated with the Hermitian holomorphic line bundle $(L, \|\cdot\|)$ over the compact complex manifold $X$ endowed with a volume form $dV_X$ at the inverse temperature $\beta/q$. By ref. [27], Theorem 5.7 (and ref. [27], Lemma 5.1) its one-point correlations measures $\nu_{N,\beta}^{(1)}$ converge as $N \to \infty$, in the weak topology of measures on $X$, toward a volume form $\nu_\beta$ on $X$ of the form

$\nu_\beta = e^{\frac{\beta}{q}\varphi_\beta} dV_X$ for the unique smooth function $\varphi_\beta$ on $X$ satisfying the following PDE on $X$:

$$\frac{1}{V}(\omega_L + dd^c\varphi_\beta)^2 = e^{\frac{\beta}{q}\varphi_\beta} dV_X, \qquad (37)$$

where $\omega_L$ is the Kähler form defined by the curvature of the metric $\|\cdot\|$ on $L$, locally expressed as $\omega_L = dd^c\phi_0(z)$. Using that $\omega_L = dd^c \log(r_0)^2/3q$ a direct calculation now reveals that the radial function $r_\beta$ on $Y$ defined by

$$r_\beta := \left(\frac{\mu_\beta}{dV_0}\right)^{1/6\beta} r_0, \qquad (38)$$

satisfies the PDE (26). This proves the convergence in item 1 of Theorem B with $\mu_\beta$ the volume form in $\mathcal{P}(M)^\xi$ corresponding to $\nu_\beta$ in $\mathcal{P}(X)$. Similarly, applying [27], Corollary 5.8 then proves the convergence of $r_{N,\beta}$ and $g_{N,\beta}$ toward $r_\beta$ and $g_\beta$, respectively. Moreover, if $(M, \xi)$ admits a Sasaki–Einstein metric, then it corresponds to a Kähler–Einstein metric $\omega_X$ on $X$. Thus, as shown in Step 2 of the proof of ref. [29], Theorem 7.9, the PDE (37) on $X$ admits a unique solution $\varphi_\beta$ for any $\beta > -1$ and $\varphi_\beta$ defines a real-analytic family converging to the Kähler potential $\varphi_{-1}$ of a Kähler–Einstein metric on $X$, as $\beta \to -1$. When rephrased in terms of $r_\beta$ and $g_{M,\beta}$ this concludes the proof of Theorem B in the regular case. Finally, in the case when $\xi$ is quasi-regular one can proceed essentially as in the regular case, using that in this case, the quotient $X$ is a Fano orbifold, so that the role of $K_X$ is now played by the orbifold canonical line bundle of $X$.

This concludes the proof of Theorem B.

**Specialization to the toric case.** In this section, we specialize our proposal to the case of a toric quiver gauge theory. As is well-known in this case the corresponding complex cone $Y$ is a toric affine Gorenstein variety. As shown in refs. [31],[32] $(Y, \xi)$ admits a conical Calabi–Yau metric $g_Y$ iff $\xi$ is the unique minimizer of the volume functional $V(\xi)$ on the space of normalized Reeb vectors, introduced in ref. [33]. Equivalently, from the gauge theory point of view, this means that the $U(1)_R$-symmetry induced by $\xi$ maximizes the $a$-central charge. We explain how $g_Y$ emerges from our proposal using a tropicalization procedure, which renders the proposal provably convergent and computationally feasible. It also applies to irregular Reeb vector fields.

Let $Y$ be a 3-dimensional normal affine toric variety. This means that $Y$ is a normal affine variety endowed with the holomorphic action of the 3-dimensional complex torus 3 with an open dense orbit, where $T_{\mathbb{C}} := (\mathbb{C}^*)^3$ denotes the complexification of the compact torus $T := U(1)^3$. Accordingly, we can identify $T_{\mathbb{C}}$ with an open subset of $Y$ and view $Y$ as an $T_{\mathbb{C}}$–equivariant compactification of $T_{\mathbb{C}}$. Denote by $y_0$ the unique point in $Y$ which is fixed under $T_{\mathbb{C}}$ and by $z = (z_1, z_2, z_3)$ the holomorphic coordinates on $T_{\mathbb{C}}$. Since it requires no extra effort we will allow $y_0$ to be a non-isolated singularity, following[32].

The ring $\mathcal{R}(Y)$, consisting of all holomorphic polynomials on $Y$, splits with respect to the $T_{\mathbb{C}}$-action on $Y$:

$$\mathcal{R}(Y) = \bigoplus_{p \in \mathcal{C}^* \cap \mathbb{Z}^3} \mathbb{C}z^p, \quad z^p := z_1^{p_1} z_2^{p_1} z_3^{p_3}, \qquad (39)$$

where $\mathcal{C}^*$ is the "moment polytope" of the affine toric variety $Y$. $\mathcal{C}^*$ can be represented as the convex cone in $(\mathbb{R}^3)^*$ whose dual is a convex cone $\mathcal{C} \subset \mathbb{R}^3$. To simplify the notation we will identify the dual $(\mathbb{R}^3)^*$ with $\mathbb{R}^3$ in the usual way. The Reeb vector fields $\xi$ on $Y$ may be identified with vectors $\boldsymbol{\xi}$ in $\mathbb{R}^3$ lying in the interior of the cone $\mathcal{C}$. Denote by $\lambda_k$ the corresponding weights $\langle \boldsymbol{\xi}, \boldsymbol{p} \rangle$ as $\boldsymbol{p}$ ranges over $\mathcal{C}^* \cap \mathbb{Z}^3$, ordered so that $\lambda_1 < \lambda_2... <$. We can thus represent the corresponding weight spaces of $\mathcal{R}(Y)$ as

$$H_{\lambda_k}(Y) = \bigoplus_{i=1}^{N_k} \mathbb{C}z^p, \qquad (40)$$

where we have enumerated the lattice points $\boldsymbol{p}_1, ..., \boldsymbol{p}_{N_k}$ in the 2-dimensional convex polytope $P_k$ defined as the intersection of the convex cone $\mathcal{C}^*$ with the hyperplane $\{\langle \boldsymbol{\xi}, \cdot \rangle = \lambda_k\}$:

$$P_k = \lambda_k P_\xi \cap \mathbb{Z}^3, \quad P_\xi := \{\mathcal{C}^* \cap \langle \boldsymbol{\xi}, \cdot \rangle = 1\}, \qquad (41)$$

(note that the polytope $P_\xi$ is denoted by $\Delta$ in ref. [9] and called the characteristic polytope). Now assume that $Y$ is Gorenstein and denote by $\Omega$ the $T_{\mathbb{C}}$–equivariant holomorphic top-form on $Y - \{y^0\}$. On $T_{\mathbb{C}} \subset Y$, we can express

$$\Omega = z^l \Omega_0, \quad \Omega_0 := \frac{dz_1}{z_1} \wedge \frac{dz_2}{z_2} \wedge \frac{dz_3}{z_3} \qquad (42)$$

for some $l \in \mathbb{Z}^3$, where $\Omega_0$ is the standard $T_{\mathbb{C}}$–invariant volume form on $T_{\mathbb{C}}$. The condition that $\Omega$ is homogeneous of degree 3 under the Reeb field $\xi$ translates into the condition $\langle l, \xi \rangle = 3$. For example, when $Y = \mathbb{C}^3$ with the standard Reeb vector $\boldsymbol{\xi} = (1, 1, 1)$, the cone $\mathcal{C}^*$ is the positive octant in $\mathbb{R}^3$ and $P_\xi$ bounds the unit-simplex. We will also briefly discuss the case of the conifold below.

We now specialize our proposal to the toric case. First, note that using the bases $z^{p_1}, ..., z^{p_{N_k}}$ in $H_{\lambda_k}(Y)$ the corresponding Slater determinant $\Psi_{\det}$ may be

represented as the following holomorphic function on $T_{\mathbb{C}}^{N_k} \subset Y^{N_k}$:

$$\Psi_{\det}(z_1,...,z_{N_k}) = \sum_{\sigma \in S_{N_k}} (-1)^{|\sigma|} z^{p_{\sigma(1)}} \cdots z^{p_{\sigma(N_k)}} \quad (43)$$

Accordingly, on the open dense subset $T_{\mathbb{C}}^N$ of $Y^N$ we can, using formula (42), express

$$|\Psi_{N_k}|^2 (\Omega \wedge \overline{\Omega})^{\otimes N_k} = \rho_{N_k}(z_1,...,z_{N_k})(\Omega_0 \wedge \overline{\Omega}_0)^{\otimes N_k}, \quad (44)$$

where

$$\rho_{N_k}(z_1,...,z_{N_k}) := \left| \sum_{\sigma \in S_{N_k}} (-1)^{|\sigma|} z^{q_{\sigma(1)}} \cdots z^{q_{\sigma(N_k)}} \right|^{-2\frac{3}{\lambda_k}}. \quad (45)$$

Here we have defined $q_i := p_i/\lambda_k - l/3$, $i = 1,...,N_k$, corresponding to the discrete points of the scaled and shifted polytope $Q_k$ defined as

$$Q_k := P_k/\lambda_k - l/3 \subset \mathbb{R}^3 \cap (\mathbb{Z}/\lambda_k)^3$$
$$Q_\xi := P_\xi - \ell/3, \quad (46)$$

where $Q_\xi$ is the limit of $Q_k$ when $k \to \infty$.

From a computational point of view, this explicit expression for $\rho_{N_k}(z_1,...,z_{N_k})$ is still rather challenging to work with directly. But the construction of a Sasaki–Einstein metric can be simplified by further leveraging the toric structure.

To see this, first recall that, in general, the group $\mathcal{G}(Y,\xi)$ of all biholomorphisms of a complex cone $Y$, commuting with the Reeb vector field $\xi$ and homotopic to the identity, acts transitively on the space of conical Calabi–Yau metrics $g_Y$[20]. In particular, the toric case $\mathcal{G}(Y,\xi)$ contains the group $T_{\mathbb{C}}$ and thus $g_Y$ is not uniquely determined, but can be taken to be $T$-invariant. The density $\rho_{N_k}(z_1,...,z_{N_k})$, on the other hand, is not $T^{N_k}$-invariant. This is to be expected as the large $N$-limit should encapsulate all conical Calabi–Yau metrics on $Y$—not only the $T$-invariant ones. In order to directly extract a $T$-invariant conical Calabi–Yau metric $g_Y$ from the large $N$-limit we can modify the density $\rho_{N_k}$ on $Y^{N_k}$ to render it $T^{N_k}$-invariant. This can be achieved in various ways, but from a computational point of view, the most efficient modification appears to replace the density $\rho_{N_k}(z_1,...,z_{N_k})$ with its tropicalization:

$$\rho_{\text{trop}}^{(N_k)}(z_1,...,z_{N_k}) := \left( \max_{\sigma \in S_{N_k}} \left| z_1^{q_{\sigma(1)}} \cdots z_{N_k}^{q_{\sigma(N_k)}} \right|^2 \right)^{-3}, \quad (47)$$

where, in particular, the sum over $S_{N_k}$ has been replaced with a maximum. In terms of the logarithmic real coordinates

$$\boldsymbol{x} = (x_1, x_2, x_3) := \left( \log(|z_1|^2), \log(|z_2|^2), \log(|z_3|^2) \right) \in \mathbb{R}^3 \quad (48)$$

this means that

$$\rho_{\text{trop}}^{(N_k)}(z_1,...,z_{N_k}) := e^{-NE_{\text{trop}}^{(N_k)}(\boldsymbol{x}_1,...,\boldsymbol{x}_{N_k})}, \quad (49)$$

where $E_{\text{trop}}^{(N_k)}(\boldsymbol{x}_1,...,\boldsymbol{x}_{N_k})$ denotes the following symmetric piece-wise affine convex function on $(\mathbb{R}^3)^{N_k}$:

$$E_{\text{trop}}^{(N_k)}(\boldsymbol{x}_1,...,\boldsymbol{x}_{N_k}) = \frac{3}{N_k} \max_{\sigma \in S_{N_k}} \left( \langle \boldsymbol{x}_1, \boldsymbol{q}_{\sigma(1)} \rangle + \cdots \right.$$
$$\left. \cdots + \langle \boldsymbol{x}_{N_k}, \boldsymbol{q}_{\sigma(N_k)} \rangle \right). \quad (50)$$

Hence, the corresponding $T^{N_k}$-invariant "tropicalized" measure on $Y^{N_k}$ may be expressed as follows

$$\rho_{\text{trop}}^{(N_k)}(z_1,...,z_{N_k})(\Omega \wedge \overline{\Omega})^{\otimes N_k} = e^{-N_k E_{\text{trop}}^{(N_k)}(\boldsymbol{x}_1,...,\boldsymbol{x}_{N_k})}(\text{d}\boldsymbol{x})^{\otimes N_k} \otimes (\text{d}\boldsymbol{\theta})^{\otimes N_k}, \quad (51)$$

with $\text{d}\boldsymbol{\theta}$ denoting the $T$-invariant probability measure on $T$. Contracting $\Omega \wedge \overline{\Omega}$ with the dilation vector field $\delta := -J\xi$, as before, thus yields a $T^{N_k}$-invariant measure on $M^{N_k}$. By performing a linear change of coordinates on $Y$, we may assume that dilatation on $Y$ corresponds to translations in the $x_3$-variable in $\mathbb{R}^3$. The contraction in question thus corresponds to replacing $\mathbb{R}^3$ in Eq. (51) by $\mathbb{R}^2$. However, the integral over $M^{N_k}$ given by the corresponding normalizing constant always diverges due to the diagonal action of the residual symmetry group $T_{\mathbb{C}}/T$. This action has the effect of translating the center of mass in $\mathbb{R}^2$ of a configuration $(\boldsymbol{x}_1,...,\boldsymbol{x}_{N_k}) \in (\mathbb{R}^2)^{N_k}$. The remedy is to break the symmetry in question. This can be achieved by introducing a background radial function (as in Theorem B). But from a computational point of view the most efficient way is to simply impose the constraint that the center of mass in $\mathbb{R}^2$ of $(\boldsymbol{x}_1,...,\boldsymbol{x}_{N_k})$ vanishes, i.e., that $\boldsymbol{x}_1 + ... + \boldsymbol{x}_{N_k}$ vanishes. Finally, since the number $N_k$ may scale as $o(k^2)$, unless $\xi$ is quasi-regular, we replace $N_k$ with any positive integer $N$ and take $q_1,...,q_N$ to be any sequence of points in the polytope $Q_\xi$ with the property that

$$\lim_{N \to \infty} \frac{1}{N} \sum_{i=1}^{N} \delta_{q_i} = \nu_{Q_\xi} \quad (52)$$

where $\nu_{Q_\xi}$ is the Euclidean measure on $Q_\xi$ normalized to have unit total mass. In summary, the tropicalized probability measure thus corresponds to the following Boltzmann–Gibbs measure on $(\mathbb{R}^2)^N$:

$$\mu_{\text{trop}}^{(N)} := \frac{1}{Z_{\text{trop}}^{(N)}} e^{-NE_{\text{trop}}^{(N)}(\boldsymbol{x}_1,...,\boldsymbol{x}_N)}(\text{d}\boldsymbol{x})^{\otimes N}, \quad (53)$$

with

$$Z_{\text{trop}}^{(N)} := \int e^{-NE_{\text{trop}}^{(N)}(\boldsymbol{x}_1,...,\boldsymbol{x}_N)} \text{d}\boldsymbol{x}^{\otimes N} \quad (54)$$

together with the constraint of vanishing center of mass.

We then have the following result.

*Theorem C:* $Z_{\text{trop}}^{(N)} < \infty$ for $N$ sufficiently large iff $(M, \xi)$ admits a conical Sasaki–Einstein metric (i.e., iff $\xi$ minimizes the volume functional $V(\xi)$ on the Reeb cone). Moreover, if this is the case then the law of the empirical measure $\frac{1}{N}\sum_{i=1}^{N} \delta_{\boldsymbol{x}_i}$ on the ensemble $\left((\mathbb{R}^2)^N, \mu_{\text{trop}}^{(N)}\right)$ converges in law, as $N \to \infty$, toward a volume form $\mu_{SE}$ on $\mathbb{R}^2$; the normalized volume form of the unique $T$-invariant Sasaki–Einstein metric on $(M, \xi)$ with vanishing center of mass (when expressed in real logarithmic coordinates on $\mathbb{R}^2$).

The probability measure $\mu_{SE}$ on $\mathbb{R}^2$ may be expressed as

$$\mu_{SE} = e^{-3\phi} \text{d}\boldsymbol{x}, \quad (55)$$

where the function $\phi$ on $\mathbb{R}^2$ corresponds to the Kähler potential $r^2$ of the corresponding $T$-invariant conical Calabi–Yau metric on $Y$, i.e., $r^2 = e^{\phi(x) + \langle l, x \rangle/3}$. Theorem C thus provides evidence for Conjecture A. The advantage of the tropicalized setup is that the corresponding energy $E_{\text{trop}}^{(N)}(\boldsymbol{x}_1,...,\boldsymbol{x}_N)$ is continuous. The theorem can be shown using results in ref. [34], where a closely related tropical approach to Kähler–Einstein metrics on toric Fano varieties was introduced. Details will appear elsewhere.

For example, in the homogeneous cases of $\mathbb{C}^3$ and the conifold it is well-known that the polytope $Q_\xi$ is a translation of the two-dimensional simplex or the unit-square, respectively. In these cases, the function $\phi(x)$ is given explicitly, modulo an additive constant, by $\phi(x) = \log(1 + e^{x_1} + e^{x_2}) - x_1/3 - x_2/3$ for $\mathbb{C}^3$ and $\phi(x) = \log(e^{-x_1/2} + e^{x_1/2}) + \log(e^{-x_2/2} + e^{+x_2/2})$ for the conifold.

Similarly, in the case when $Y$ is $\mathbb{C}^2$ the polytope $Q_\xi$ is equal to $[-1/2, 1/2]$ and the corresponding one-point correlations can be computed explicitly for any finite $N$. The result is a polynomial of degree $N$ in $e^{-|x|}$, converging, as $N \to \infty$, toward $e^{-2\phi(x)}$, where $\phi(x) = \log(e^{-x/2} + e^{x/2}) - C$. Indeed, in this case, the corresponding tropical energy $E_{\text{trop}}^{(N)}$ turns out to coincide with the energy of a self-gravitating system in 1D (with a mean-field scaling), to which the exact results in ref. [35] apply.

In general, however, the solution $\phi$ can not be explicitly computed. An important feature of Theorem C is that it provides an efficient way of obtaining numerical approximations to the solution $\phi$ and thus to the Sasaki–Einstein metric on $M$, using Hamiltonian Monte-Carlo. The starting point is the observation that the gradient of the energy $NE_{\text{trop}}^{(N)}$ appearing in the Boltzmann–Gibbs measure (51) is, for a generic configuration $(\boldsymbol{x}_1,...,\boldsymbol{x}_N) \in \mathbb{R}^{2N}$, precisely the discrete optimal transport map matching the points $(\boldsymbol{x}_1,...,\boldsymbol{x}_N)$ with the fixed points $\boldsymbol{q}_1,...,\boldsymbol{q}_N$ in $Q_\xi$[36]. Thanks to the last years rapid developments of scalable optimal transport solvers this allows one to numerically compute the gradient in nearly $\mathcal{O}(N)$ operations (efficiently implemented on GPU-hardware[37,38]). Moreover, since $E_{\text{trop}}^{(N)}$ is convex Hamiltonian Monte-Carlo should merely require $\mathcal{O}(N^{1/4})$ gradient evaluations[39], suggesting that the total running time of the simulation nearly scales as $\mathcal{O}(N^{1+1/4})$.

Monte-Carlo techniques have previously been applied in ref. [40] to the vacuum moduli space $Y^N/S_N$ in the case $Y = \mathbb{C}^3$, but using a different $N$-particle BPS-wave function (with a Metropolis algorithm).

We leave the implementation of our simulation scheme for the future.

## Data availability

Data sharing is not applicable to this article since no data sets were generated or analyzed during the current study.

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

## Acknowledgements

We are grateful to David Berenstein, Amihay Hanany, and James Sparks for helpful discussions and correspondence. The work of R.J.B. was supported by the Swedish Research Council (grant no. 11253043), the Wallenberg Foundation (grant no. 11253045), and the Göran Gustafsson prize (grant no. 11253042). T.C.C. was supported by NSF CAREER grant (no. DMS-1944952), and an Alfred P. Sloan Fellowship. D.P. was supported by the Swedish Research Council (grant. no 2018-04760), and the Wallenberg AI, Autonomous Systems and Software Program (WASP) funded by the Wallenberg Foundation (grant no. 2020.0173).

## Author contributions

R.J.B., T.C.C. and D.P. have all contributed equally to this work. Authors are listed alphabetically according to surnames.

## Funding

## Competing interests

The authors declare no competing interests.
