## [Peer Review File · Nature Communications]

Reviewers' Comments:

Reviewer #1:

Remarks to the Author:

This paper studies a class of examples of the AdS/CFT (Anti-de Sitter/Conformal Field Theory) correspondence, i.e the duality between type IIB string theories on $AdS_5 \times M$ ($M =$ Sasaki-Einstein manifolds) and four-dimensional $\mathcal{N}=1$ supersymmetric quiver gauge theories.

There are two main claims in the paper: (1) The metric on M , Sasaki-Einstein manifolds, can be reconstructed from what the authors dubbed, a "canonical quantum BPS state" in quiver gauge theories. In a more popular term, this is an "emergence" of a space(-time) from non-gravitational dual CFT. (2) The "canonical quantum BPS state" has an interpretation as a bound state of dual giant gravitons in AdS5, or alternatively, a coherent state of giant gravitons in M .

In this class of the AdS/CFT correspondence, for the working of the duality, it is a requirement that the quantum moduli space of vacua of a quiver gauge theory is Y^N/S_N where Y is a Calabi-Yau cone over M and N is the rank of gauge groups. In this sense, the emergence of the internal space M from the dual gauge theory is a built-in structure of AdS/CFT. A novelty in this paper is a claim that the metric on M can be explicitly reconstructed from a quantity, an S_N scale, and R -symmetry invariant, built out of BPS states (or operators).

My comment on claim (1) is that from their construction of the "canonical quantum BPS state", it is not clear if the metric on M has been or can be really obtained. It would be helpful if the authors can provide (A) a concrete application of their construction to the case of $N = 4$ SYM as a benchmark example and (B) a concrete example of the $Y^{p,q}$ case for which the metric is known, in order to demonstrate how their proposal actually works.

On claim (2), it does not seem to me that convincing evidence has been provided to support their claim. Once again, it would be helpful if the authors elaborate on their claim, for example, by taking $N = 4$ SYM as a prototypical example. In this case, it is well-established that (dual) giant gravitons correspond to Schur polynomial operators of Z_i ($i = 1, 2, 3$) specified by Young diagrams. A bound state of dual giant gravitons is a view of the diagram as a collection of rows, whereas a bound state of giant gravitons is a view of the diagram as a collection of columns. These two views are the "duality" of giant gravitons. I am not sure if this can have a coherent state interpretation at all.

Reviewer #2:

Remarks to the Author:

The paper "Emergent Sasaki-Einstein geometry and AdS/CFT" is a novel development in the understanding of geometry in string theory setups related to the AdS/CFT correspondence.

To start the analysis, in supersymmetric gauge theories that arise from string theory, it is often the case that one encounters metrics that are cones over Sasaki-Einstein metrics.

These are a special class of non-compact Calabi-Yau geometries.

Mathematically, these are geometries with a special radial variable. The Sasaki-Einstein geometry M , can be taken to be the geometric locus at a fixed value of the radius

In these setups, one would like to determine the metrics on the cones from the dual CFT directly. This has only been possible so far in cases with maximal symmetry.

In this paper, the authors make a new construction that gives rise to a Sasaki-Einstein metric construction from first principles.

The idea is to define a holomorphic "wave function" on the space of a set of N particles on M for a sequence of values of N that can grow arbitrarily large. From this setup, one can construct a probability measure on

N copies of M and by averaging one produces a probability measure on a single copy of M . The authors find that in the large N limit of this sequence, the measure converges.

The existence of this measure, using other theorems in math, then shows a construction that gives rise to a Sasaki-Einstein metric.

This is a very interesting development in the area. It uses novel algebraic geometry techniques and mathematics, plus a physics ansatz construction in a very interesting way.

There are a couple of points that need a bit more clarification: I had a very hard time figuring out how singular the proposed wave function is. The authors state that they have a proof of integrability of the measure in a forthcoming paper. It would be good to have at least one example showing how bad the singularity is (transversely to the singularity).

I was also confused about the dual giant gravitons prescription for interpretation of the singularity. It is true that the system goes singular when all points belong to the zero locus of ANY holomorphic function. This makes the points weakly attract to the singularity, but not too much (otherwise the transverse measure would not be integrable). This one that is picked defines a giant graviton "baryon" type configuration. But because the singularity does not discriminate these, the giant graviton is also averaged over all possibilities somehow.

I think that overall, the novelty of the construction alone would justify publication. It is definitely of a lot of interest to both physicists and mathematicians.

Reviewer #3:

Remarks to the Author:

This paper proposes to get an emergent Sasaki-Einstein metric from a dual CFT by applying a procedure that is described in detail.

The treatment is mostly done on the geometry side and very little is done on the dual gauge theory. The authors somehow assume that a dual description is known, but this is not always correct. The simplest example is C^3 and it is discussed in several places. Here a dual CFT is known. The next example is the set of singularities of equation 13. This set is a very problematic set. A dual field theory is known for the case $p=q=2$ and not for the other cases, as far as I can tell. Hence it is not correct to assume that there exists a dual theory. One first needs to work it out. Let us take the case where a dual field theory is known. The conifold theory. It would be extremely useful to have explicit expressions that test the proposal. If the computations are too hard to evaluate, then this is probably the main difficult aspect of the paper. One has a proposal, but evaluating it on examples, even simple ones, is a major task. So no matter how the proposal is nice, it becomes prohibitive due to lack of computability.

I will be happy to have another look once the example of the conifold is worked out.

Response to Referee 1

“Emergent Sasaki-Einstein metrics and AdS/CFT”

Robert Berman, Tristan Collins, Daniel Persson

– submission to *Nature Communications* –

Dear Referee,

Thank you very much for the nice and valuable feedback on our paper. We have created a new and substantial revision of the paper, taking into account your, and the other referees’, comments and questions. The new additions/reformulations are color coded in the new version. Below we offer a point-by-point detailed response to your comments, indicating where in the new draft your queries are addressed.

“My comment on claim (1) is that from their construction of the “canonical quantum BPS state”, it is not clear if the metric on M has been or can be really obtained. It would be helpful if the authors can provide (A) a concrete application of their construction to the case of $\mathcal{N} = 4$ SYM as a benchmark example and (B) a concrete example of the $Y^{p,q}$ case for which the metric is known, in order to demonstrate how their proposal actually works.”

In response to this we have added section III.B of the paper [light blue text] in which we consider the toric case in detail (see also the red text in section I). The metrics corresponding to the cones \mathbb{C}^3 , the conifold or $Y^{p,q}$ all fall into this class. We first provide an explicit form of the canonical BPS-state in terms of a holomorphic function of the complex coordinates on $(\mathbb{C}^*)^3$. We then discuss the main new result in the toric setup, Theorem C, which provides an efficient way of obtaining numerical approximations to any toric Sasaki-Einstein metric. Finally, we briefly sketch a simulation scheme using so called “Hamiltonian Monte-Carlo” [blue text at the end of section III.B]. The implementation of this simulation scheme will be given in a future publication.

“On claim (2), it does not seem to me that convincing evidence has been provided to support their claim. Once again, it would be helpful if the authors elaborate on their claim, for example, by taking $\mathcal{N} = 4$ SYM as a prototypical example. In this case, it is well-established that (dual) giant gravitons correspond to Schur polynomial operators of Z_i ($i = 1, 2, 3$) specified by Young diagrams. A bound state of dual giant gravitons is a view of the diagram as a collection of rows, whereas a bound state of giant gravitons is a view of the diagram as a collection of columns. These two views are the “duality” of giant gravitons. I am not sure if this can have a coherent state interpretation at all.”

We agree that more evidence to support claim (2) should be provided. Given that the new section III.B is a major addition to the paper, we decided to remove the discussion of giant gravitons from the present paper. We will come back to giant gravitons in a future publication, taking the referee’s questions into account. See the text in magenta at the end of section I for a comment on this.

Sincerely,

Robert Berman

Tristan Collins

Daniel Persson

Response to Referee 2

“Emergent Sasaki-Einstein metrics and AdS/CFT”

Robert Berman, Tristan Collins, Daniel Persson

– submission to *Nature Communications* –

Dear Referee,

Thank you very much for the nice and valuable feedback on our paper. We have created a new and substantial revision of the paper, taking into account your, and the other referees’, comments and questions. The new additions/reformulations are color coded in the new version. Below we offer a detailed response to your comments, indicating where in the new draft your queries are addressed.

“There are a couple of points that need a bit more clarification: I had a very hard time figuring out how singular the proposed wave function is. The authors state that they have a proof of integrability of the measure in a forthcoming paper. It would be good to have at least one example showing how bad the singularity is (transversely to the singularity).”

In order to address this point we have added a discussion of this at the end of section II.D. The new discussion is included in a subsection entitled “Finiteness properties of the normalizing constant” [red text]. It begins after eq. (28) and continues until the very end of section II.C.

Sincerely,

Robert Berman

Tristan Collins

Daniel Persson

Response to Referee 3

“Emergent Sasaki-Einstein metrics and AdS/CFT”

Robert Berman, Tristan Collins, Daniel Persson

– submission to *Nature Communications* –

Dear Referee,

Thank you very much for the nice and valuable feedback on our paper. We have created a new and substantial revision of the paper, taking into account your, and the other referees’, comments and questions. The new additions/reformulations are color coded in the new version. Below we offer a point-by-point detailed response to your comments, indicating where in the new draft your queries are addressed.

“The treatment is mostly done on the geometry side and very little is done on the dual gauge theory. The authors somehow assume that a dual description is known, but this is not always correct. The simplest example is \mathbb{C}^3 and it is discussed in several places. Here a dual CFT is known. The next example is the set of singularities of equation 13. This set is a very problematic set. A dual field theory is known for the case $p = q = 2$ and not for the other cases, as far as I can tell. Hence it is not correct to assume that there exists a dual theory. One first needs to work it out.”

We agree that constructing the dual field theory is in general a complicated task. Our proposal, however, only requires that the moduli space of vacua of the field theory, given by the complex cone Y , is known. We have clarified this in the paper [see green text at the end of section IIA], emphasizing that we do not assume that a complete field theory is known for all Sasaki-Einstein metrics. Rather, our proposal can be viewed as the statement that whenever a field theory exists, and has a moduli space given by Y (or, rather, Y^N/S_N), we describe how the Sasaki-Einstein metric emerges from a canonical BPS-state on Y^N/S_N .

“Let us take the case where a dual field theory is known. The conifold theory. It would be extremely useful to have explicit expressions that test the proposal. If the computations are too hard to evaluate, then this is probably the main difficult aspect of the paper. One has a proposal, but evaluating it on examples, even simple ones, is a major task. So no matter how the proposal is nice, it becomes prohibitive due to lack of computability. I will be happy to have another look once the example of the conifold is worked out.”

In response to this we have added section III.B of the paper [light blue text] in which we consider the toric case in detail (see also the red text in section I). The metrics corresponding to the cones \mathbb{C}^3 , the conifold or $Y^{p,q}$ all fall into this class. We first provide an explicit form of the canonical BPS-state in terms of a holomorphic function of the complex coordinates on $(\mathbb{C}^*)^3$. We then discuss the main new result in the toric setup, Theorem C, which provides an efficient way of obtaining numerical approximations to any toric Sasaki-Einstein metric.

Finally, we briefly sketch a simulation scheme using so called “Hamiltonian Monte-Carlo” [blue text at the end of section III.B]. The implementation of this simulation scheme will be given in a future publication.

Sincerely,

Robert Berman

Tristan Collins

Daniel Persson

Reviewers' Comments:

Reviewer #1:

Remarks to the Author:

I do appreciate the effort the authors made in order to clarify the questions raised in the referee report. However, even with the elaborations in the revised version, the conjecture made in this article, that is the main result, is still short of evidence. The crucial technical points are relegated to their forthcoming paper Ref.[25]. So it is appropriate to wait for the forthcoming results.

The main claim involves an introduction of (1) the "canonical" BPS "state" as dubbed by the authors and (2) the 1-point correlation measure of a "canonical ensemble" constructed from (1). Their conjecture is that the latter (2) in a large N limit yields the volume form of a Sasaki-Einstein (SE) manifold M which is the base of a Calabi-Yau (CY) cone Y . If true, the Kahler potential (or the radial function squared) can be constructed and the Kahler metric can be found.

As an attempt to prove their conjecture, (2) is generalized to a one-parameter family of the measure. It can be thought of as introducing an inverse temperature β . That is where the connection to a canonical ensemble arises. To obtain (2) which is of their interest, the inverse temperature β needs to be analytically continued to -1 .

From physics viewpoints, first of all, it is hard to gain any intuitions for their proposed construction of the Kahler metric: The "canonical" BPS "state" is somewhat a natural quantity to consider in the sense that it is basically a determinant operator for a fixed R -charge sector. (As a side remark, this is why it smells like giant gravitons.) This "state" is then used to compose a scale invariant measure that is a necessary condition to obtain the volume form of a SE manifold which is scale invariant. However, unless proved, a layman like myself cannot help wondering why this is expected to work.

Second, more importantly, as a canonical ensemble, their system seems to be unstable because $\beta = -1$ would correspond to particles in an attractive potential. So the particles tend to clump together and collapse. When β is positive, since it would correspond to a repulsive potential, the system seems to be well-defined. That is why the introduction of β was necessary for the authors. However, from physics viewpoints, it is hard to imagine how the analytic continuation from repulsion, crossing $\beta = 0$, to attraction can be smooth and justified.

Nevertheless, the claim is that all these criticisms can be rigorously and mathematically addressed and justified. However, it is relegated to the forthcoming paper Ref.[25] (including the mean field approximation that I could not quite understand).

As requested in the previous referee report, even though the author added an example of a toric quiver gauge theory, it does not really clarify how this construction works and why it can be useful. This example faced its own technical difficulties and a modification to the original claim was needed. So it did not really get to the bottom.

To summarize the report, what lacks to me as one of the physicist referees is clearer evidence for how their construction might be working and a more intuitive understanding of their proposal.

Reviewer #2:

Remarks to the Author:

The revised paper is much improved. It helped to see the example of the sphere in more detail. Also, the topological arguments are useful to understand better the arguments in examples.

I think the paper is very interesting and contributes to an area of interest both to Physicists and Mathematicians.

The paper should be published in its revised version.

Reviewer #3:
None

2nd response to Referee 1

“Emergent Sasaki-Einstein metrics and AdS/CFT”

Robert Berman, Tristan Collins, Daniel Persson

– submission to *Nature Communications* –

Dear Referee,

Thank you very much for the additional feedback on our paper. We have created a new revision of the paper, taking into account your comments. The new additions/reformulations are color coded in the new version. Below we offer a point-by-point detailed response to your comments, indicating where in the new draft your queries are addressed.

“I do appreciate the effort the authors made in order to clarify the questions raised in the referee report. However, even with the elaborations in the revised version, the conjecture made in this article, that is the main result, is still short of evidence. The crucial technical points are relegated to their forthcoming paper Ref.[25]. So it is appropriate to wait for the forthcoming results. ”

The main criticism is the lack of a rigorous mathematical proof of our main claim. Indeed, we had originally planned to postpone this to a forthcoming paper, but we agree with the referee that this is a drawback of the present paper. In the new version we have therefore addressed this by including the complete mathematical proof our claim. The proof is given in section III.A (blue color). We hope that this addresses the concern of the referee. In the process we have also slightly shortened section III.B and reformulated the statement of Theorem B in order to make it more explicit and clear (in magenta).

Sincerely,

Robert Berman

Tristan Collins

Daniel Persson

Reviewers' Comments:

Reviewer #1:

Remarks to the Author:

I appreciate the authors' response and their efforts to strengthen their claims. This paper may have interests and appeals to mathematicians but not to physicists who work in the field of AdS/CFT or holography and who try to understand gravity from gauge theory. I must admit that I do not think this paper is suitable for publication in Nature Communications.

I repeat the following questions raised in the previous report:

(1) As a canonical ensemble, their system seems to be unstable because $\beta = -1$ would correspond to particles in an attractive potential. So the particles tend to clump together and collapse. When β is positive, since it would correspond to a repulsive potential, the system seems to be well-defined. That is why the introduction of β was necessary for the authors. However, from physics viewpoints, it is hard to imagine how the analytic continuation from repulsion, crossing $\beta = 0$, to attraction can be smooth and justified.

(2) What lacks to me as one of the physicist referees is clearer evidence for how their construction might be working and a more intuitive understanding of their proposal.

Nature communications is for all areas of the biological, health, physical, chemical and Earth sciences. As far as I can see, there are neither any physical intuitions nor examples, for physicists, based on very concrete quiver gauge theories that show the benefits of authors' claims.

3rd response to Referee 1

“Emergent Sasaki-Einstein metrics and AdS/CFT”

Robert Berman, Tristan Collins, Daniel Persson

– submission to *Nature Communications* –

Dear Referee,

Thank you very much for the additional feedback on our paper. Below we offer a detailed response to your comments.

Reply to Comment 1

Consider a general canonical ensemble of N particles on a compact Riemannian manifold M with interaction energy $E^{(N)}$, at inverse temperature β , represented by the Gibbs measure

$$\frac{1}{\mathcal{Z}_N(\beta)} e^{-\beta E^{(N)}} dV^{\otimes N}, \quad \mathcal{Z}_N(\beta) := \int_{X^N} e^{-\beta E^{(N)}} dV^{\otimes N}.$$

There are different ways of defining stability for such a system, but the most general definition simply demands that the partition function be finite, $\mathcal{Z}_N(\beta) < \infty$. Indeed, this is precisely what is needed for the Gibbs measure to be a well-defined probability measure and thus for the system to be defined as a statistical ensemble. In our situation the interaction $E^{(N)}$ is repulsive in the sense that $E^{(N)}(x_1, \dots, x_N) \rightarrow \infty$ as two particle locations merge. Thus, when $\beta < 0$ the “effective interaction” $\beta E^{(N)}$ is attractive, i.e. $\beta E^{(N)}(x_1, \dots, x_N) \rightarrow -\infty$ as two particles merge. This could be viewed as a sign of instability, as particles appear to clump together. But the point is that the system can still be stable in the aforementioned sense, i.e. $\mathcal{Z}_N(\beta) < \infty$.

In fact, in our case, for any Calabi-Yau cone with base M there exists a strictly positive number ϵ (which is independent of N) such that

$$\beta \geq -\epsilon \implies \mathcal{Z}_N(\beta) < \infty. \tag{1}$$

It then follows from basic calculus that the meromorphic function

$$\beta \mapsto \mathcal{Z}_N(\beta) \tag{2}$$

is analytic in a neighborhood of the strip in \mathbb{C} where the real part of β is larger than $-\epsilon$. In particular, the behavior of $\mathcal{Z}_N(\beta)$ is completely smooth as β changes sign, from $]0, \infty[$ down to $] -\epsilon, \infty[$. This may, perhaps, seem surprising. But it essentially boils down to the fact that if f is a holomorphic function in \mathbb{C}^n , then $|f|^{-2\epsilon}$ is locally integrable if ϵ is a sufficiently small number. Indeed, after resolution of singularities and separation of variables the problem is reduced to the one-variable case where $f(z)$ is a monomial.

A major challenge in our situation, however, is to prove that if M admits a Sasaki-Einstein metric, then one can take $\epsilon = 1$, since our canonical BPS state corresponds to the Gibbs ensemble above at $\beta = -1$. Our proposal in the paper is to sidestep this difficulty by

first taking β to be positive. Our main result shows that if one then lets $N \rightarrow \infty$, the large N -limit dV_β of the one point-correlation measures is a real-analytic function of β all the way down to $\beta = -1$ and dV_{-1} is a Sasaki-Einstein metric.

To sum up: the main technical problem is not really that one need to cross from positive β to negative β , but that one needs to go all the way down to $\beta = -1$. We still expect that, in fact, $\mathcal{Z}_N(-1) < \infty$ for finite N , as long as N is taken to be sufficiently large. But this problem is left for the future. We do, however, give examples in the paper of Calabi-Yau cones for which $\mathcal{Z}_N(-1) < \infty$ for all N , namely when the Calabi-Yau cone is an exceptional singularity (for example, $Y = \mathbb{C}^3/G$ for an appropriate finite subgroup of $SU(3)$, e.g. Klein's simple group of order 168).

Reply to Comment 2

It would, indeed, be great to have clearer evidence for how the construction works physically and a more intuitive understanding of our proposal. Hopefully our work can stimulate developments in this direction in the future!

Sincerely,

Robert Berman

Tristan Collins

Daniel Persson